# Factors Related to HPV Vaccination Intention among MSM in China: A Bayesian Network Model

**DOI:** 10.3390/ijerph192315532

**Published:** 2022-11-23

**Authors:** Qiao Chen, Tianyi Zhou, Xiaoni Zhong

**Affiliations:** School of Public Health, Chongqing Medical University, Chongqing 400016, China

**Keywords:** men who have sex with men (MSM), vaccination intention, human papillomavirus (HPV) vaccine, correlates, Bayesian networks (BNs)

## Abstract

(1) Background: Men who have sex with men (MSM) are at high risk of human papillomavirus (HPV) infection, and HPV vaccination is the best strategy to prevent HPV infection. Accepting HPV vaccination is an essential factor affecting vaccine promotion among MSM. We aimed to explore the factors related to HPV vaccination intention among MSM and analyze the potential relationship between these factors. (2) Methods: We adopted a nonprobability sampling method to recruit MSM volunteers. Information collection included general demographics, personal behavioral characteristics, knowledge of HPV/vaccine attitudes, and risk threat perception. Bayesian networks (BNs) were used to analyze the data statistically. (3) Results: The BNs showed that perceived HPV risk and attitudes toward vaccine promotion were directly correlated factors, whereas knowledge of HPV/vaccines, a history of HIV testing, and the number of male sexual partners in the past 6 months were indirectly correlated factors. (4) Conclusions: The results of this study illustrate that MSM have a relatively high propensity to receive HPV vaccines. The proposal that strengthening the propagation of HPV and its relevant vaccines, encouraging MSM to undergo regular corresponding tests, and improving their risk perception of HPV infection can be raised to promote HPV vaccination among MSM.

## 1. Introduction

Human papillomavirus (HPV) is a spherical DNA virus of family *Papillomaviridae* that causes squamous epithelial hyperplasia of the skin and is the most common sexually transmitted virus in the world [1]. HPV infection can cause cancers, such as cervical, oropharyngeal, anal, and vulvar cancers [2,3,4]. It is the second most common source of cancer transmission after *Helicobacter pylori* [5]. Most HPV infections do not show obvious clinical symptoms and can clear spontaneously, but persistent HPV infection can lead to various benign and malignant lesions [6]. Sexual transmission is the main transmission route of HPV infection, and approximately 80% of sexually active men or women will be infected with certain types of HPV during their lifetime [1,7]. In women, HPV infection may lead to a range of gynecological cancers, such as cervical, vaginal, and vulvar cancers [2,8]. In comparison, HPV infection in men has received little attention; the prevalence of HPV infection in men is high, with internationally reported infection rates ranging from 5.3% to 42.2% [9]. In China, the prevalence of HPV infection is around 10.5% [10].

The most cost-effective way to prevent HPV infection is to be vaccinated against HPV [11]. Several studies have shown that HPV vaccination is effective in protecting women from HPV in the cervical, vulvar, and anal regions [12,13]. Vaccination also plays a remarkable role in preventing HPV infection in men in the anal, penile, and oral regions [14,15]. The World Health Organization (WHO) recommends that the HPV vaccine should be included in national immunization programs. However, the current global recommendation for HPV vaccination is still mainly for women and is offered in two ways: self-funded and free. Some developed countries, such as the USA and Australia, already offer free nationwide HPV vaccination programs for young men [16]. In Mainland China, the HPV vaccine was approved for marketing in 2016, but the target population is limited to women due to its high price and low supply [17]. In fact, the HPV vaccination program for women has shown a group effect, and heterosexual men can also benefit from it. However, men who have sex with men (MSM) do not benefit from herd immunity through vaccination in women due to the particularity of their sexual partners [18]. Moreover, evidence from several studies showed that the HPV vaccine can remarkably benefit MSM and reduce HPV-related infections [19,20].

In addition to national policy and vaccine availability, the willingness to receive the HPV vaccine is also an essential factor influencing the use of the vaccine among the MSM population. Therefore, a comprehensive analysis of the factors associated with HPV vaccination intention among the MSM population is vital to increase HPV vaccination coverage among the MSM population. Previous studies have used logistic regression based on independent conditions to explore factors associated with vaccination intention and have used odds ratios (ORs) to reflect the degree of association [21,22,23]. In reality, factors are often interdependent, and relationships may have a complex network structure. Bayesian networks (BNs) are an artificial intelligence method that does not require strict statistical assumptions [24]. The potential relationship between multiple factors is reflected by constructing a directed acyclic graph, and the strength of the association is reflected by a conditional probability distribution table [25,26]. In addition, BNs can use the state of known nodes (i.e., factors) to infer the probability of unknown nodes (i.e., vaccination intentions), which may be a more flexible means of understanding the likelihood of HPV vaccination among MSM populations. Given the appealing nature of BNs, we used BNs to model the factors associated with HPV vaccination intention among MSM populations and identify potential relationships between these factors, informing and suggesting the promotion of HPV vaccination in MSM populations. Therefore, this study is of great public health importance. 

## 2. Materials and Methods

### 2.1. Subjects of Study

This study is a cross-sectional study. Nonprobability sampling was used to recruit MSM respondents from non-governmental organizations owing to the specificity of this population. Relevant data were collected by means of a web-based questionnaire distributed via WeChat, and respondents voluntarily chose to fill out the web-based questionnaire after understanding the purpose of the study and providing informed consent. The participants who completed the questionnaire were given a reward of 10 RMB. The study was approved by the Ethics Committee of Chongqing Medical University (2019001, 28 May 2019).

### 2.2. Inclusion and Exclusion Criteria

A total of 1477 MSM were recruited, excluding 422 who took less than 2 min to fill in the form, 69 who were younger than 14 years or older than 55 years, and 17 who had missing or incorrect answers in the logic check. A total of 969 MSM were finally included in the study.

### 2.3. Questionnaire Content

The web-based questionnaire for this study consisted of three main sections focusing on the current status of HPV vaccination intention and its associated factors among the MSM population. The first section included general demographic characteristics, such as the age, education level, and monthly disposable income of the individual. The second section investigated the behavioral characteristics of the MSM population, such as the number of male and female partners in the last 6 months and whether they had been actively tested for human immunodeficiency virus (HIV) or HPV, etc. Finally, the third section investigated knowledge attitudes and perceived risk threat of HPV/vaccines in the MSM population, focusing on variables such as knowledge about HPV disease and vaccines, perceived HPV risk, and attitudes toward vaccine promotion. Twelve questions focused on knowledge of HPV disease and vaccines, in which a score of 10 or more indicated a high level of knowledge, and a score less than 10 indicated a medium or low level of knowledge.

### 2.4. Statistical Analysis

Statistical analysis was performed using R software (version 4.2.1, https://www.r-project.org accessed on 12 August 2022). Count data were analyzed descriptively using rates or composition ratios. The stats package in R software was used to analyze correlated factors. One-way logistic regression analysis was used for the initial screening of correlated factors, and the test level alpha was set at 0.05. The variables that were statistically significant in the one-way analysis were selected as the network nodes for BN construction, and the bnlearn package in R software was used to construct the BN model. The Tabu search algorithm was used for the structural learning of the BNs, and the great likelihood estimation method was used for the parameter learning of the BNs. Furthermore, GeNIe software (version 4.0, https://download.bayesfusion.com accessed on 15 August 2022) was used for BN model visualization and conditional probability inference.

## 3. Results

### 3.1. General Demographic Characteristics and HPV Vaccination Intention

Table 1 shows the proportions of MSM with different demographic characteristics who were willing to be vaccinated against HPV. Univariate analysis showed that MSM’s household and marital status were relevant factors for HPV vaccination. The proportions of urban and rural MSM who were willing to receive the HPV vaccine were 84.85% and 79.64%, respectively. Urban MSM were 1.432 times more likely to receive the HPV vaccine than rural MSM. Among marital status, MSM with unmarried status had a higher propensity to be vaccinated, being approximately 1.613 times more likely than those with a married status. No statistical differences in HPV vaccination intention were found among the variables of age, region, ethnicity, education level, employment status, and monthly disposable income (*p* > 0.05).

### 3.2. Behavioral Characteristics and HPV Vaccination Intention

Table 2 shows the proportions of MSM with different behavioral characteristics regarding their willingness to receive HPV vaccination. The results show that the number of male partners in the last 6 months, the number of female partners in the last 6 months, a history of HIV testing, a history of sexually transmitted disease (STD) testing, and a diagnosis of an STD in the last 6 months were all relevant factors influencing HPV vaccination. MSM who had multiple male partners in the last 6 months were 1.846 times more likely to be vaccinated than those who had no male partners. MSM who had an STD in the last 6 months were 1.732 times more likely to be vaccinated than MSM who did not have an STD. Proactive HIV testing and STD testing were also boosters of vaccination, with ORs of 2.441 and 2.162, respectively.

### 3.3. Knowledge Attitudes, Risk Threat Perception, and HPV Vaccination Intention

Table 3 shows the proportions of vaccination intention among MSM with different knowledge attitudes and risk threat perceptions. The level of HPV/vaccine knowledge, perceived HPV risk, perceived HPV threat, and attitudes to vaccine promotion were all statistically correlated with vaccination intention, but no statistical differences were found between maintaining a neutral attitude toward vaccine promotion and having an opposing attitude toward MSM vaccination. Among these factors, holding a positive attitude toward vaccine promotion was highly associated with vaccination intention (OR = 10.441, 95% CI = 3.546–32.242). MSM with high levels of HPV risk perception were 5.992 times more likely to be vaccinated compared with MSM with low levels of HPV risk perception. In terms of HPV threat perception levels, those with high levels of HPV threat perception had higher willingness to vaccinate, approximately 3.826 times higher than those with low levels of HPV threat perception. Moreover, MSM with high levels of HPV/vaccine knowledge were more willing to receive HPV vaccines than those with a medium/low level of knowledge (OR = 3.119, 95% CI = 2.180–4.528).

### 3.4. Bayesian Network Model

The statistically significant variables (*p* < 0.05) in the one-way logistic regression analysis were used as the sample information, and BNs were used to construct a model of the factors associated with the willingness to be vaccinated against HPV in the MSM population, with 14 directed edges at 12 nodes. As shown in Figure 1, the association between vaccination intention and the correlated factors in the MSM population is a complex network relationship. Attitudes toward vaccine promotion and the level of perceived HPV risk are directly correlated with HPV vaccination among the MSM population, whereas the number of male sexual partners indirectly influences the willingness of the MSM population to receive HPV vaccination through the level of perceived HPV risk. The BNs also showed that HIV testing and HPV vaccine knowledge were parent nodes of attitudes toward vaccine promotion, which indirectly influence the MSM population’s willingness to receive the HPV vaccine.

### 3.5. Conditional Probabilistic Reasoning

A remarkable advantage of BNs is that the probabilities of unknown nodes can be inferred from the probabilities of known nodes. In the developed BN model, the level of perceived HPV risk and the attitudes toward vaccine promotion are the parent nodes of vaccination intention. Supposing that an MSM has a high level of HPV risk perception, he will have a 95% probability of receiving the HPV vaccine (Figure 2). If this MSM also approves of vaccine promotion, his probability of receiving the vaccine will increase to 97% (Figure 3). The BN diagram also depicts clearly the interrelationships between the factors involved. Assuming that an MSM has little information about the vaccine and is opposed to vaccine promotion, he has a 27% probability of receiving the HPV vaccine, a 13% probability of having a high level of perceived HPV risk, a 14% probability of having been tested for HIV, and a 14% probability of having a high level of knowledge about HPV/vaccines (Figure 4).

## 4. Discussion

### 4.1. Willingness to Vaccinate

The MSM population is at high risk of HPV infection, and HPV vaccination is the most cost-effective way to prevent HPV infection. However, no vaccination program is currently available for the MSM population in Mainland China [17]. The present study showed that 82.77% of MSM were willing to receive the HPV vaccine. This percentage is higher than the previously reported willingness to vaccinate among MSM in Shenyang, China (57.7%), the internationally reported willingness to vaccinate in Greece (76.7%), and the previously reported willingness to vaccinate among the target female population recommended for HPV vaccination in China (67.25%) [27,28,29]. The substantially higher willingness to vaccinate in comparison with other studies may be due to the higher literacy level of this MSM population (73.37% of the MSM had tertiary education or higher). Previous studies found that MSM’s willingness to receive the HPV vaccine is related to their education level [30]. MSM with a high level of education have a clear understanding of the risks of HPV infection and the benefits of HPV vaccination and are thus more willing to receive HPV vaccination. Therefore, the HPV vaccine can potentially be promoted among MSM at high risk of HPV infection.

### 4.2. Relevant Factors

BNs can explore the complex network relationships between factors associated with HPV vaccination intention among MSM populations. The results of the BNs suggest that the perceived HPV risk and attitudes toward vaccine promotion are directly associated factors influencing HPV vaccination among MSM populations. These results are consistent with the results of traditional logistic regression models. We also found that HPV/vaccine knowledge, HIV testing history, and the number of male sexual partners in the past 6 months were indirectly associated factors.

The perceived level of HPV risk in the MSM population was directly correlated with HPV vaccination intention. This finding is consistent with the findings of He et al. [31]. MSM with high levels of HPV risk perception are more likely to receive HPV vaccination and expect to reduce their risk of HPV infection through vaccination. Moreover, the number of male sexual partners in the past 6 months was indirectly correlated with HPV vaccination intention. In general, MSM with a higher number of male partners are at greater risk of HPV infection and are more likely to receive the HPV vaccine.

Attitudes toward vaccine promotion are also directly correlated with HPV vaccination intention among MSM. A history of HIV testing is an indirect correlate of vaccination intention. Moreover, knowledge of HPV/vaccines is correlated with attitudes toward vaccine promotion, as it indirectly influences the intention to receive the HPV vaccine. The knowledge–attitude–behavior theory explains how a person’s knowledge directly influences his/her attitude and indirectly influences behavior through his/her attitude [32]. The theory demonstrates the correlation between knowledge about HPV/vaccines, attitudes toward vaccine promotion, and willingness to vaccinate. In the present study, MSM with low to moderate HPV/vaccine knowledge levels accounted for 50% of the total MSM population tested. Therefore, educational interventions should be implemented to improve the knowledge of HPV/vaccines among MSM. While implementing educational interventions, we should use straightforward methods to ensure that the benefits and importance of HPV vaccination are understood regardless of the level of education, to facilitate the roll-out of the HPV vaccine among the MSM population.

This study has used BNs to analyze the factors associated with HPV vaccination intention among MSM. The findings are consistent with previous studies showing that BNs can accurately identify relevant factors. Moreover, the BNs graphically describe the complex network probability mechanism of HPV vaccination intention among MSM, through which we can identify the relevant factors influencing the intention to receive HPV vaccination among MSM and explore the potential correlation between these factors. Although traditional multivariate logistic regression models can also explore the factors associated with HPV vaccination intention among MSM, they cannot describe the interrelationships between factors.

However, this study has some limitations. First, this study is cross-sectional, and the directed edges between nodes in the BN graph can only represent conditional dependencies between nodes. They cannot be considered causal relationships. Second, the sampling method was nonprobability sampling; therefore, the results of the study may have some bias because of the specificity of the MSM population. Third, this study did not ask the MSM about the price of the HPV vaccine and the institution or organization where they would like to be vaccinated against HPV. The cost of vaccination and the mode of access may also be essential factors affecting the actual vaccination rate among the MSM population [33,34]. In the future, we will further study the impact of vaccination costs and acquisition methods on HPV vaccination among the MSM population.

## 5. Conclusions

Men who have sex with men (MSM) are a high-risk group for HPV infection. In this study, 82.77% of MSM were willing to be vaccinated against HPV and had a high propensity to be vaccinated. Therefore, the authorities should consider including the MSM population in the national vaccination program. In addition, using a Bayesian network model, we identified HPV risk perception and vaccine promotion attitudes as direct correlates of HPV vaccination among MSM. Moreover, HPV/vaccine knowledge, HIV testing history, and the number of male sexual partners in the past 6 months are indirect correlates. Therefore, we should strengthen the promotion of knowledge about HPV and HPV vaccines, encourage MSM to take the initiative to be tested regularly for HIV/HPV, and increase the risk perception of HPV infection to aid the promotion of HPV vaccination among MSM.

## Figures and Tables

**Figure 1 ijerph-19-15532-f001:**
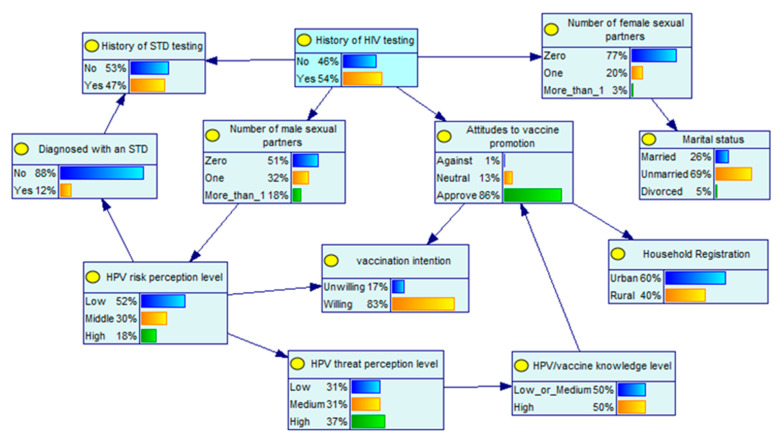
Bayesian network simulation diagram.

**Figure 2 ijerph-19-15532-f002:**
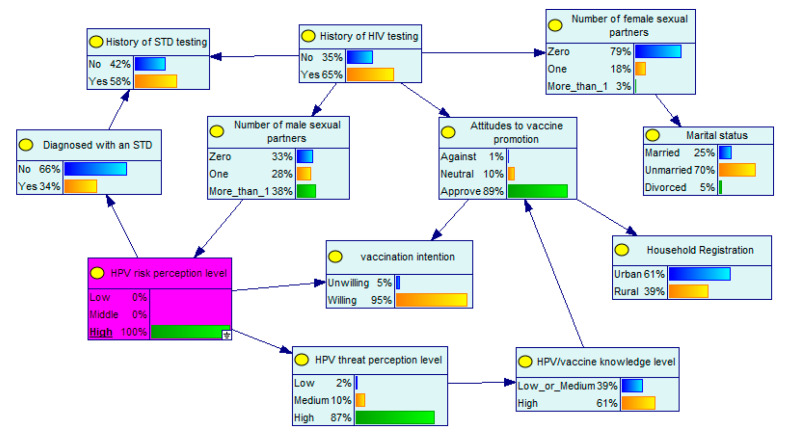
Bayesian network with known node probabilities I.

**Figure 3 ijerph-19-15532-f003:**
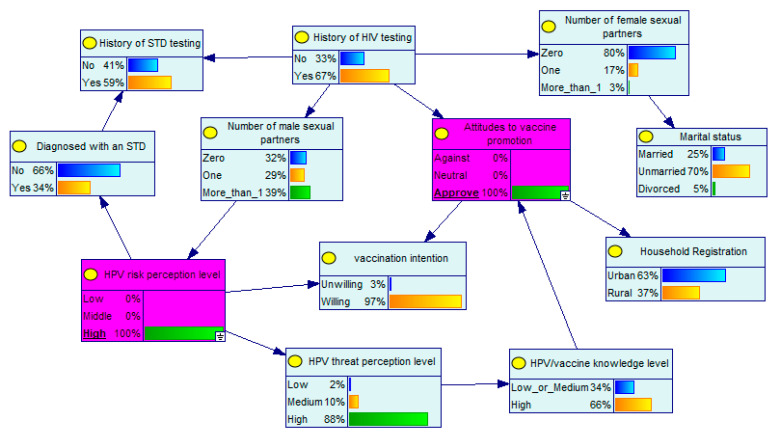
Bayesian network with known node probabilities II.

**Figure 4 ijerph-19-15532-f004:**
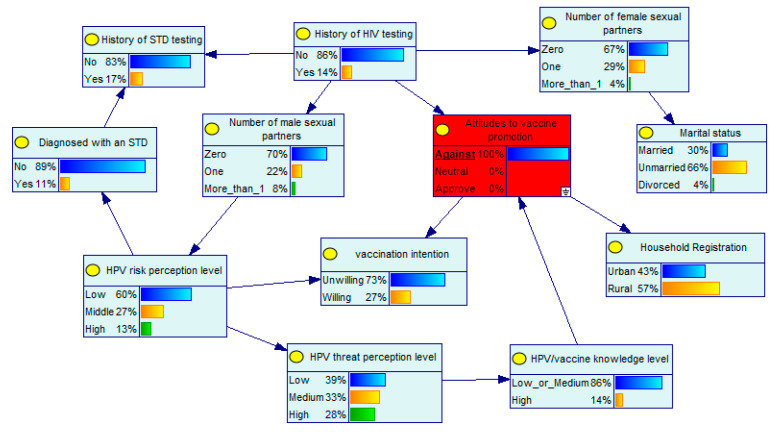
Bayesian network with known node probabilities III.

**Table 1 ijerph-19-15532-t001:** Comparison of differences in HPV vaccination willingness among different demographic characteristics in MSM.

Variable	Cases	Willing to Be Vaccinated		OR (95% CI)	*p* Value
		N	%		
**Total**	969	802	82.77		
**Age**					
14–18	33	27	81.82	Reference	
18–35	719	598	83.17	1.098 (0.403, 2.547)	0.839
35–55	217	177	81.57	0.983 (0.349, 2.399)	0.972
**Household Registration**					
Rural	388	309	79.64	Reference	
Urban	581	493	84.85	1.432 (1.023, 2.003)	0.036
**Region**					
Developed	254	206	81.10	Reference	
Moderately developed	193	153	79.27	0.891 (0.558, 1.429)	0.630
Less developed	522	443	84.87	1.307 (0.876, 1.933)	0.184
**Ethnicity**					
Han nationality	933	733	82.85	Reference	
Other nationalities	36	29	80.56	0.858 (0.390, 2.159)	0.721
**Education level**					
Illiterate/semi-literate	9	7	77.78	Reference	
Primary	10	7	70.00	0.667 (0.070, 5.291)	0.702
Junior high	71	53	74.65	0.841 (0.118, 3.871)	0.838
High/secondary school	168	138	82.14	1.314 (0.190, 5.765)	0.741
Junior college	234	185	79.06	1.079 (0.157, 4.632)	0.926
College and above	477	412	86.37	1.811 (0.266, 7.687)	0.465
**Employment status**					
Employed	707	591	83.59	Reference	
Student	168	139	82.74	0.941 (0.609, 1.493)	0.789
Unemployed	94	72	76.60	0.642 (0.389, 1.098)	0.094
**Marital status**					
Married	250	194	77.60	Reference	
Unmarried	672	570	84.82	1.613 (1.116, 2.315)	0.010
Divorced	47	38	80.85	1.219 (0.577, 2.822)	0.621
**Monthly disposable income**					
1000 RMB or less	76	58	76.32	Reference	
1001–3000 RMB	222	182	81.98	1.412 (0.740, 2.623)	0.283
3001–5000 RMB	250	212	84.80	1.731 (0.906, 3.223)	0.089
5001–10000 RMB	296	242	81.76	1.391 (0.744, 2.513)	0.286
10000 RMB or more	125	108	86.40	1.972 (0.943, 4.144)	0.071

**Table 2 ijerph-19-15532-t002:** Comparison of differences in HPV vaccination willingness among different behavioral characteristics in MSM.

Variable	Cases	Willing to Be Vaccinated		OR (95%CI)	*p* Value
		N	%		
**Number of male sexual partners** **(last 6 months)**					
0	490	374	76.33	Reference	
1	306	272	88.89	2.481 (1.659, 3.797)	<0.001
>1	173	156	90.17	2.846 (1.697, 5.051)	<0.001
**Number of female sexual partners** **(last 6 months)**					
0	703	628	84.52	Reference	
1	197	155	78.68	0.676 (0.458, 1.011)	0.052
>1	29	19	65.52	0.348 (0.161, 0.797)	0.009
**History of STD testing**					
No	514	400	77.82	Reference	
Yes	455	402	88.35	2.162 (1.525, 3.099)	<0.001
**History of HIV testing**					
No	448	341	76.12	Reference	
Yes	521	461	88.48	2.411 (1.712, 3.422)	<0.001
**Diagnosed with an STD** **(last 6 months)**					
No	852	694	81.46	Reference	
Yes	117	108	92.31	2.732 (1.429, 5.910)	0.005

**Notes.** STD: sexually transmitted disease; HIV: human immunodeficiency virus.

**Table 3 ijerph-19-15532-t003:** Comparison of differences in HPV vaccination willingness among MSM with different knowledge, attitude, and risk/threat perception levels.

Variable	Cases	Willing to Be Vaccinated		OR (95% CI)	*p* Value
		N	%		
**HPV/vaccine knowledge level**					
Medium/low level	481	361	75.05	Reference	
High level	488	441	90.37	3.119 (2.180, 4.528)	<0.001
**HPV risk perception level**					
Low level	504	390	77.38	Reference	
Medium level	293	248	84.64	1.611 (1.109, 2.374)	0.020
High level	172	164	95.35	5.992 (3.040, 13.608)	<0.001
**HPV threat perception level**					
Low level	305	224	73.44	Reference	
Medium level	305	250	81.97	1.644 (1.119, 2.429)	0.012
High level	359	328	91.36	3.826 (2.470, 6.060)	<0.001
**Attitudes to vaccine promotion**					
Against	14	6	42.86	Reference	
Neutral	127	62	48.82	1.272 (0.419, 4.061)	0.6723
Approve	828	734	88.65	10.411 (3.546, 32.242)	<0.001

## Data Availability

The data presented in this study are available upon request from the corresponding authors. The data are not publicly available as they contain sensitive personal details.

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
