# Peer review of "Factors Related to HPV Vaccination Intention among MSM in China: A Bayesian Network Model"

_ijerph, 2022, doi:10.3390/ijerph192315532_

Round 1
Reviewer 1 Report
Some revisions are needed before which I puted on comments.

Reviewer 2 Report
The authors have to be congratulated for the development of Bayesian network models regarding the probability of vaccination in MSM. Currently, there is a high interest in the topic of this paper, as male vaccination is gaining popularity. Moreover, data have been published that MSM do not benefit from herd immunity of female vaccination compared to men who have sex with vaccinated women. Additionally, Head and Neck squamous cell carcinoma (HNSCC) is currently the most frequent HPV-related tumor with a high prevalence in men. Despite the subject of this article is of high relevance, I feel like the data presented here are insufficient for publication. I have listed some major comments:
- In the introduction at line 37 the authors state that HPV infection in women is caused by cervical and vulvar cancer. However, I suggest to mention it the other way round: HPV infection causes cervical cancer and a big part of vulvar cancers (also penile cancers and so forth).
- At line 45 the authors mention that HPV vaccination can prevent men from penile cancer. However, to our knowledge there are no data on the benefits of HPV vaccination in penile cancer. The only thing we can say is that penile cancer is related to HPV in 50.8% of cases, so these men possibly may benefit from vaccination.
- I suggest to do some multivariate analysis instead of univariate analysis because all characteristics would definitely influence each other. Completely different results could emerge if doing multivariate analysis.
- At line 159: what do the authors mean with a high risk of self-infection compared to low levels of HPV/vaccine knowledge? This is a quite confusing sentence.
- The HPV/vaccine knowledge level is significantly correlated with the willingness to be vaccinated. I was wondering how the authors made the difference between knowledge level? It is quite subjective if they just assessed the knowledge level by filling in high or low in a questionnaire.
- It would be useful to include a paragraph on vaccination costs. Maybe more men are willing to be vaccinated if these vaccines are paid by the government. This factor would also be correlated with employment status.
Round 2
Reviewer 2 Report
I thank the authors for their additions and modifications in the manuscript. The readability is already much better. However I still have some minor suggestions
- Line 33: "some HPV" --> certain types of HPV is maybe a better way to say this.
- Line 36: HPV infection has been described to cause cervical, vulvar and other gynAEcological cancers. In my opinion it is more accurate to mention all these gynaecological cancer than just say 'some other cancers'.
- Line 40-42. I think this sentence is quite stigmatizing to MSM. Please mention an objective argument as well: MSM do not benefit from herd immunity through vaccination in women.
- line 58-60: this sentence is grammatically not correct.
I suggest for the whole paper to let it check by a native English speaker.
- Thank you for clarification in line 102-104 about the knowledge level. It might be a suggestion to split up low and moderate as well, based on another treshold and mention 2 OR in line 164.
- Thank you for the inclusion of the cost-willingness paragraph.
